# Immunogenicity and Safety of One versus Two Doses of Quadrivalent Inactivated Influenza Vaccine (IIV4) in Vaccine-Unprimed Children and One Dose of IIV4 in Vaccine-Primed Children Aged 3–8 Years

**DOI:** 10.3390/vaccines11101586

**Published:** 2023-10-12

**Authors:** Yunfeng Shi, Wanqi Yang, Xiaoyu Li, Kai Chu, Jianfeng Wang, Rong Tang, Li Xu, Lanshu Li, Yuansheng Hu, Chenyan Zhao, Hongxing Pan

**Affiliations:** 1Jiangsu Provincial Center for Disease Control and Prevention, Nanjing 210009,China; 2Sinovac Biotech Co., Ltd., Beijing 100085, China; 3National Institutes for Food and Drug Control, Beijing 102629, China

**Keywords:** inactivated influenza vaccine, immunogenicity, safety, two-dose regimen in children

## Abstract

Two doses of the inactivated influenza vaccine (IIV) are generally recommended for children under 9 years old. This study assessed the necessity for a second dose of quadrivalent IIV (IIV4) in children aged 3–8 years. In this randomized, open-label, paralleled-controlled study, 400 children aged 3–8 years who were vaccine-unprimed were randomly assigned at a 1:1 ratio to receive a two-dose (Group 1) or one-dose (Group 2) regimen of IIV4, and 200 who were vaccine-primed received one dose of IIV4 (Group 3). A serum sample was collected before and 28 days after the last dose to determine the hemagglutination inhibition (HI) antibody level. Adverse events were collected within 28 days after each dose. One-dose or two-doses of IIV4 were well tolerated and safe in children aged 3–8 years, and no serious adverse events related to the vaccine were reported. The seroconversion rates (SCRs) of HI antibody ranged from 61.86% to 95.86%, and the post-vaccination seroprotection rates (SPRs) were all >70% in three groups against the four virus strains. The two-dose regimen in vaccine-unprimed participants (Group 1) achieved similar SPRs in comparison with the one-dose in the vaccine-primed group (Group 3), and the SPRs in Group 1 and Group 3 were higher in vaccine-unprimed participants of the one-dose regimen (Group 2). The present study supports the recommendations of a two-dose regimen for IIV4 use in children aged 3–8 years.

## 1. Introduction

Children are susceptible to seasonal influenza, with an estimated annual global attack rate of 20–30% in children compared with 5–10% in adults [1]. Influenza virus infections occurring in children may lead to severe and fatal symptoms. A population-based study conducted in China showed that 69% of influenza-related severe acute respiratory infection (SARI) hospitalizations occurred in children under the age of 5 [1]. A modeling study on global influenza-related deaths estimated that 9243–105,690 children under the age of 5 die from influenza-related respiratory diseases every year in 92 countries with high rates of mortality due to respiratory infection [2].

To reduce the risk of influenza virus infection and the burden of severe complications, annual vaccination is the most effective approach [3]. Although regulatory authorities and academia have reached consensus about the importance of influenza vaccination for children, the optimal dose regimen remains controversial. The 2022–2023 Technical Guidelines for Influenza Vaccination from the China Center for Disease Control and Prevention (CDC) recommend that: regarding the inactivated influenza vaccine (IIV), children 6 months through 8 years who are receiving influenza vaccine for the first time should receive two doses administered ≥4 weeks apart; children who have received one dose or more during the years 2021–2022 or before should receive only one dose [4]. The American Academy of Pediatrics (AAP), however, recommends that this pediatric group receive two doses of influenza vaccine in children who have not previously received ≥2 doses of trivalent or quadrivalent inactivated influenza vaccine before 1 July 2022 [5]. In an attempt to provide a reference for optimizing the dose regimen of the influenza vaccine in these children, we conducted this phase IV clinical trial.

A quadrivalent inactivated influenza vaccine (IIV4), manufactured by Sinovac Biotech Co. Ltd., has been licensed for influenza prevention in China in 2020 and is indicated from 3 years of age. The pivotal phase I/III clinical trial of the IIV4 demonstrated satisfactory immunogenicity and safety results [6]. However, only the one-dose regimen of IIV4 was evaluated, and the immune response in children with various vaccination histories was not studied separately [6]. Therefore, it is necessary to conduct additional studies on the immunogenicity and safety of two-dose or one-dose IIV4 for vaccine-unprimed children, as well as one-dose IIV4 for vaccine-primed children. The objective of the current study is to explore the optimal influenza immunization strategy for children aged 3–8 years.

## 2. Methods

### 2.1. Study Design

This phase IV, randomized, open-label, paralleled-controlled study was conducted in Huai’an County, Jiangsu, China (ClinicalTrial.gov identifier: NCT04997239) between 10 October 2021 and 20 December 2021. Participants with no influenza vaccination or one-dose influenza vaccination experience were considered to be vaccine-unprimed; participants who had cumulatively received two or more doses of influenza vaccine before were considered to be vaccine-primed. The protocol was approved by the Ethics Committee of the Jiangsu Provincial Center for Disease Control and Prevention, and this study was conducted in accordance with the Declaration of Helsinki, Good Clinical Practice (GCP), and all applicable laws and regulations. Written informed consent was obtained from parents/guardians before any study-related procedures were performed.

### 2.2. Study Population

Eligible participants were healthy children aged 3–8 years. The key exclusion criteria included: (1) received seasonal influenza vaccines in the 2021–2022 influenza season or a history of seasonal influenza virus infection in the past six months; (2) axillary temperature >37 °C, or acute diseases within 7 days before vaccination; (3) history of severe allergies to any vaccine components; (4) immunocompromised; (5) a known coagulation disorder; (6) received immunoglobulin or blood products within 3 months, or received immunosuppressants or hormones within 6 months; (7) received live attenuated vaccines within 14 days, or subunit vaccines or inactivated vaccines within 7 days; (8) had any other factors that were considered inappropriate for clinical trial by researchers.

### 2.3. Randomization

A total of 600 participants were enrolled, including 400 vaccine-unprimed participants who were randomized at a 1:1 ratio into two groups (Group 1 and Group 2) and 200 vaccine-primed participants in Group 3. Participants in Groups 1 and 2 were assigned a random number generated by SAS software 9.4 (SAS Institute, Inc., Cary, NC, USA) in the order in which they were enrolled and then administered the corresponding vaccination.

### 2.4. Vaccines and Vaccination Schedule

Participants in Group 1 received two doses of IIV4 on Day 0 and Day 28. Participants in Groups 2 and 3 received one dose of IIV4 on Day 0. All vaccines used for this trial were manufactured by Sinovac Biotech Co., Ltd. Each 0.5 mL dose contained 15 mcg of hemagglutinin (HA) from the four influenza strains recommended for the 2021–22 flu season by WHO for egged-based quadrivalent vaccines: A/Victoria/2570/2019 (H1N1)pdm09; A/Cambodia/e0826360/2020 (H3N2); B/Washington/02/2019 (B/Victoria lineage); B/Phuket/3073/2013 (B/Yamagata lineage) [7]. Vaccines were administered intramuscularly in the deltoid area.

### 2.5. Sample Size

This study was an exploratory study on immunization schedules, and the sample size was not based on statistical power calculations. A total of 200 participants per group were planned to be enrolled.

### 2.6. Immunogenicity Evaluation

Blood samples were collected on days 0 and 28 (window, Day 28 + 7) after the last dose. Antibody titers were measured using a hemagglutination inhibition (HI) assay performed at the China National Institutes for Food and Drug Control. Non-specific inhibitors were eliminated using receptor-destroying enzyme (cholera filtrate, Sigma-Aldrich (Shanghai) Trading Co., Ltd., Shanghai, China) at 37 °C for 16–18 h, and spontaneous agglutinins were adsorbed with Rooster red blood cells. Starting at 1:10, serum samples were 2-fold diluted for 10 times and incubated with an influenza virus solution of 4 hemagglutination units/25 μL. The highest serum dilution at which hemagglutination was completely inhibited was reported as the hemagglutination inhibition titer.

The primary objective was to compare the immunogenicity of three immunization scenarios: one-dose or two-dose IIV4 in vaccine-unprimed children, and one-dose of IIV4 in vaccine-primed children. The immunogenicity endpoints included seroprotection rate (SPR), seroconversion rate (SCR), and geometric mean titers (GMT) of HI antibody 28 days after full-course vaccination against each of the four viral strains. The SPR was defined as the percentage of participants with an HI titer ≥ 1:40, which is considered a surrogate endpoint of protection by regulatory authorities [8,9]. The SCR was defined as the percentage of participants with either a pre-vaccination HI titer < 1:10 and a post-vaccination HI antibody titer ≥ 1:40, or a pre-vaccination HI antibody titer ≥ 1:10 and a minimum 4-fold rise in post-vaccination HI antibody titer [8].

### 2.7. Safety Evaluation

Immediate adverse reactions (ARs) were observed at least 30 min after vaccination. Parents/guardians used the dairy card to record the occurrence and severity of any solicited local or systemic adverse events (AEs). The solicited local symptoms included pain, induration, swelling, erythema, rash, and pruritus. The requested systemic symptoms included fever, allergic reaction, diarrhea, decreased appetite, vomiting, nausea, myalgia, headache, cough, and fatigue. Unsolicited adverse events and serious adverse events (SAEs) were recorded within 28 days after each dose. The grading standard for adverse reactions was based on the “Guidelines for grading standard of adverse events in clinical trials of preventive vaccines” issued by the China National Medical Products Administration (NMPA) [10]. The causal relationship between AE and vaccination was determined by the investigators.

### 2.8. Statistical Analysis

Statistical analyses used SAS Version 9.4 (SAS Institute, Inc., Cary, NC, USA). Immunogenicity was assessed in the per-protocol set (PPS), which was defined as all eligible participants who were vaccinated as per protocol requirements, had pre-vaccination and post-vaccination HI titers, and did not have any laboratory-confirmed influenza illness, prohibited medications, or protocol deviations assessed as potentially affecting immunogenicity results. Safety was assessed in the overall safety set (SS), which included all participants who received at least one dose of this study vaccine and had evaluable follow-up safety data.

SPRs of the four viral strains were primary indicators. SCRs and GMTs were reference indicators. The two-sided 95% CI of the SPRs and SCRs were calculated by using Clopper-Pearson; the Pearson χ^2^ test or Fisher exact test was used to compare the difference among groups; and the Miettinen-Nurminen method was used to calculate the rate difference and two-sided 95% CI. The log-transformed GMTs were compared using the T-test method.

## 3. Results

### 3.1. Participants Disposition and Baseline Characteristics

A total of 600 participants were randomized in this study, with 200 participants in each group (Figure 1). All participants received at least one dose of the vaccine and were enrolled in the safety set (SS). A total of 49 participants discontinued from this study; the main reason was voluntary withdrawal (participants indicated unwillingness to continue in the study or participants made the decision to discontinue in this study for any personal reasons other than SAE/AE). Three participants (two in Group 1 and one in Group 2) refused blood sample collection post-vaccination. One participant has previously received two influenza vaccines and was mistakenly enrolled in Group 2. Therefore, a total of 547 participants entered the per protocol set (PPS), with 169 in Group 1, 184 in Group 2, and 194 in Group 3 (Figure 1). No participants discontinued the trial due to AEs. The basic characteristics were similar across groups (Table 1). The mean age of the 600 children at the time of the first vaccination was 5.8 years, and 49% were male. All participants in Group 3 had previously received two or more influenza vaccines. Approximately 33% of participants in Group 1 and 29.5% of participants in Group 2 had previously received one dose of the influenza vaccine.

### 3.2. Immunogenicity

For all groups, the pre-vaccination SPRs and GMTs were higher for A strains than for B strains. Participants in Group 3 had overall higher pre-vaccination SPRs and GMTs compared with participants in Group 1 and Group 2. Prior to vaccination, SPRs were less than 70% in both Group 1 and Group 2 against any of the four strains, while SPR reached 70% in Group 3 only against H3N2 (Table 2).

Both one-dose and two-dose IIV4 in vaccine-unprimed children, as well as one-dose IIV4 in vaccine-primed children, elicited strong immune responses against the four strains. The SCRs of HI antibody ranged from 61.86% to 95.86%, and the SPRs were all >70% in the three groups against the four virus strains, well meeting the influenza vaccine criteria formulated by EMA [9]. Overall, the post-vaccination SCRs in Group 1 and Group 2 were higher than those in Group 3. The post-vaccination GMTs of IIV4 against the four strains ranged from 87.20 to 852.84 (Table 2).

When comparing Groups 1 and 2, we found that Group 1 showed significantly higher SPRs against all four strains. The GMT ratios (Group 1/Group 2) were similar for each of the four strains, ranging from 0.94 to 1.14. Group 1 had significantly higher SCRs against H3N2 (89.35% vs. 77.17%, *p* = 0.0023) and BY strains (88.17% vs. 74.46%, *p* = 0.0010), compared with Group 2 (Table 3). The SPRs in Group 1 and Group 3 were similar against all four strains. The GMT ratios (Group 1/Group 3) ranged from 0.67 to 1.45 for the four strains. Compared with Group 3, Group 1 demonstrated significantly higher GMT against H1N1 (*p* < 0.0001), H3N2 (*p* < 0.0001), and BV (*p* = 0.0284) strains and significantly higher SCR against H1N1 (*p* = 0.0022), H3N2 (*p* < 0.0001), and BY (*p* = 0.0034) strains (Table 3). When comparing Group 2 and Group 3, we found that Group 2 had significantly lower SPRs against all four strains. The GMT ratios (Group 2/Group 3) ranged from 0.66 to 1.31 for the four strains. Compared with Group 3, Group 2 had significantly higher GMT against H1N1 (*p* < 0.0001) and H3N2 (*p* < 0.0001) and significantly higher SCR against H3N2 (Table 3).

According to subgroup analysis, the SPRs, SCRs, and GMTs elicited by a one- or two-dose regimen of IIV4 demonstrated no significant difference against all four strains for participants who had received one shot of influenza vaccine before enrollment (Table 4). In an analysis by prior vaccination status, a one-dose regimen of IIV4 induced significant lower SCRs against A strains in children who had received two or more doses of influenza vaccination, compared with children who had received one-dose historical vaccination or none at all. One dose of IIV4 generates significantly lower SPRs and GMTs against B strains in children with no previous dose, as compared to those with at least one previous dose (Table 5).

### 3.3. Safety

The influenza vaccine was well tolerated in all three groups. In the overall safety population (n = 600), the incidence of vaccine-related adverse events (Adverse reactions, ARs) in Group 1 after the first dose, Group 1 after the second dose, Group 2, and Group 3 was 11.5% (23/200), 3.72% (7/188), 12.5% (25/200), and 13% (26/200), respectively. After one dose of vaccination, no significant difference in adverse reaction incidence was noted across the three groups. Most ARs were mild or moderate in intensity. Only one episode of AR in grade 3 was reported by a participant in Group 2, with the symptom of pyrexia. Most ARs were general disorders and administration site conditions in terms of system organ class (SOC) and vaccine site erythema in terms of preferred term (PT), according to the Medical Dictionary for Regulatory Activities (MedDRA) coding principle. In Group 1, a markedly lower incidence of ARs was detected after the second dose compared with the first dose. During the clinical trial, no SAEs related to vaccination occurred (Table 6).

## 4. Discussion

In this study, both one-dose and two-dose schedules of IIV4 were able to elicit strong immune responses in vaccine-unprimed (defined in this study as no more than one-dose vaccination history) children aged 3–8 years, with SPRs >70% against all the four strains. However, significantly higher protective responses were induced by two doses rather than one dose of IIV4 in vaccine-unprimed children, especially for B strains (approximately 4% increase in SPR for H1N1 strain, 6% increase in SPR for H3N2 strain, 12% increase in SPR for BY strain, 11% increase in SPR for BV strain). A published immunogenicity study also reported significantly higher SPR when vaccinated with two doses of trivalent inactivated influenza vaccine (TIV) as compared to one dose in children <9 years receiving TIV for the first time [10]. We also found that the SPRs of a one-dose regimen in participants who had received two or more doses were comparable to those of a two-dose regimen in vaccine-unprimed participants. A study in Hong Kong reported that the vaccine effectiveness induced by a one- or two-dose vaccination regimen was 73% and 31%, respectively, in children aged 6 months through 9 years receiving influenza vaccination for the first time [11]. Another effectiveness study in Japan revealed that both one- and two-dose regimens of IIV significantly reduced cases of any influenza among children aged 1–12 years; however, only the two-dose regimen was significantly effective in preventing influenza B in certain influenza seasons [12].

Our data provided additional evidence that two doses of IIV4 vaccines developed by Sinovac are well tolerated in children aged 3–8 years; no SAEs or unexpected adverse reactions were discovered in the two-dose regimen compared with the previously reported single-dose regimen in the phase 3 clinical trial [6]. The above evidence supports the recommendation of a two-dose regimen of IIV in vaccine-unprimed children under 9 years old, and IIV4 developed by Sinovac could be a safe and effective choice for this vaccination regimen.

As previously introduced, minor differences existed in the 2022–2023 influenza vaccination guidelines from China and USA authorities [4,5]. Regarding children 6 months through 8 years, a one-dose regimen is recommended if they received one-dose vaccination against influenza during 2021–2022 or earlier, whereas a two-dose regimen is recommended if they received one-dose vaccination before July 2022 [5]. According to the subgroup analysis in this study, a one- or two-dose regimen of IIV4 induced equally high SPRs against all four strains in participants who had a one-dose influenza vaccination history; a one-dose regimen of IIV4 also induced comparable SPRs against all four strains in participants who had received a one- or multiple-dose influenza vaccination before. Therefore, we have reason to believe that this study children could be well protected by one dose of IIV4 if they had received at least one dose of vaccination against influenza.

Several reports have suggested that repeated vaccination might attenuate the vaccine’s effectiveness [13]. A meta-analysis funded by WHO and the US National Institutes of Health suggested that, although vaccination in the previous year attenuates vaccine effectiveness, vaccination in two consecutive years provides better protection than does no vaccination [14]. WHO also claims that vaccination in the current and prior seasons affords better protection than not being vaccinated or being vaccinated in the prior season only [15]. A study in China reported that repeated vaccination with IIV4 induced similar or enhanced immune protection as compared with single vaccination in first-time vaccinees [16]. In this study, we report significantly lower GMTs against B strains and significantly lower SCR against the H3N2 strain, 28 days after one dose of IIV4, in vaccine-unprimed children than in vaccine-primed children. It has not been shown that repeated vaccination attenuates the immune response to IIV4.

This study has several limitations. First, the sample size in this study is determined based on the basic sample size requirements for the phase II study from Chinese Regulators, but not strictly calculated based on any statistical test. Thus, *p* values and 95% CIs were provided just for reference. Second, the vaccine effectiveness of one- or two-dose regimens was not estimated in this study and might need to be further evaluated to provide a higher level of evidence for vaccination strategy development. Third, with regard to participants who had at least one dose of vaccination before this study, whether their historical vaccination occurred right in the last influenza season or not is unknown. As a result, we are not able to assess whether repeated vaccination over two consecutive seasons will attenuate the immune response to IIV4.

## 5. Conclusions

The study IIV4 demonstrated good immunogenicity and safety in children aged 3–8 years. Two-dose IIV4 in vaccine-unprimed children was able to achieve similar immune responses compared with one-dose IIV4 in vaccine-primed children. The present study supports the recommendations of a two-dose regimen for IIV4 use in children aged 3–8 years.

## Figures and Tables

**Figure 1 vaccines-11-01586-f001:**
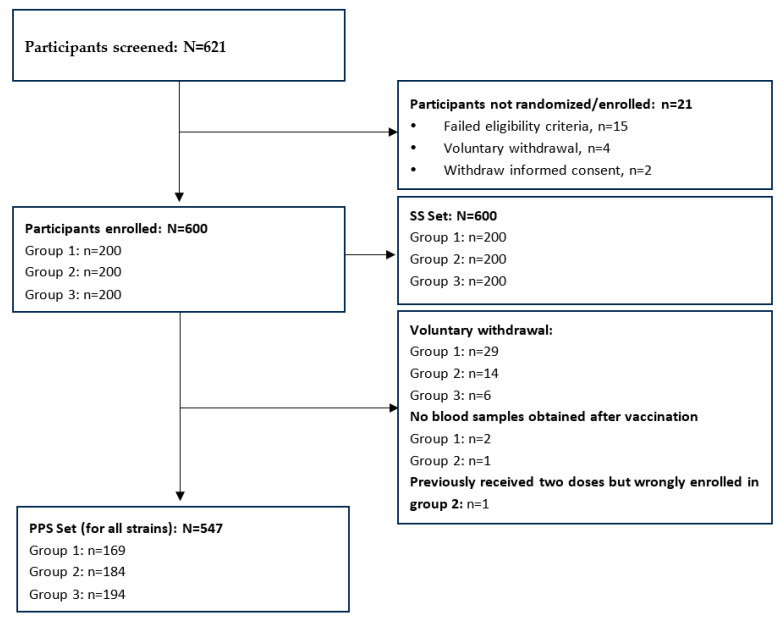
Participants disposition.

**Table 1 vaccines-11-01586-t001:** Demographics and baseline characteristics (safety set).

Characteristics	Group 1 (n = 200)	Group 2 (n = 200)	Group 3 (n = 200)	Overall (n = 600)
Age, years, mean (SD)	5.5 (1.6)	5.4 (1.7)	6.5 (1.4)	5.8 (1.7)
Gender, n (%)				
Male	102 (51.00)	94 (47.00)	98 (49.00)	294 (49.00)
Female	98 (49.00)	106 (53.00)	102 (51.00)	306 (51.00)
Han ethnicity, n (%)	200 (100.00)	200 (100.00)	199 (99.50)	599 (99.83)
Height, cm, mean (SD)	119.2 (11.0)	118.4 (11.6)	125.5 (10.3)	121.0 (11.4)
Weight, kg, mean (SD)	24.1 (7.2)	23.7 (7.7)	27.3 (7.9)	25.0 (7.7)
Previous vaccination status				
Yes, n (%)	66 (33.00)	59 (29.50)	200 (100.00)	325 (54.17)
No, n (%)	134 (67.00)	141 (70.50)	0 (0.00)	275 (45.83)
Pre-vaccination axillary temperature, °C, mean (SD)	36.23 (0.42)	36.23 (0.42)	36.33 (0.41)	36.26 (0.42)

**Table 2 vaccines-11-01586-t002:** Immunogenicity of IIV4 for each strain 28 days after the last vaccination (per-protocol set).

Time	Indicator	H1N1			H3N2			BY			BV		
		Group 1	Group 2	Group 3	Group 1	Group 2	Group 3	Group 1	Group 2	Group 3	Group 1	Group 2	Group 3
	N	169	184	194	169	184	194	169	184	194	169	184	194
Pre	SPR (%) (95% CI)	52.66 (44.85, 60.38)	49.46 (42.02, 56.91)	69.59 (62.59, 75.97)	65.68 (58.00, 72.80)	64.67 (57.30, 71.56)	77.84 (71.33, 83.47)	19.53 (13.84, 26.31)	21.00 (15.53, 27.82)	47.94 (40.73, 55.21)	13.61 (8.83, 19.72)	16.30 (11.28, 22.45)	24.23 (18.38, 30.88)
	GMT (1:) (95% CI)	30.14 (24.43, 37.18)	28.82 (23.34, 35.59)	64.33 (52.72, 78.50)	49.51 (40.12, 61.09)	53.06 (43.03, 65.43)	99.84 (82.66, 120.59)	16.36 (14.89, 17.98)	17.73 (15.93, 19.73)	31.26 (28.13, 34.73)	12.33 (10.83, 14.04)	12.87 (11.24, 14.74)	15.74 (13.76, 18.01)
Post	SCR (%) (95% CI)	95.86 (91.65, 98.32)	91.85 (86.91, 95.37)	86.60 (80.98, 91.05)	89.35 (83.69, 93.56)	77.17 (70.42, 83.03)	61.86 (54.62, 68.72)	88.17 (82.32, 92.62)	74.46 (67.52, 80.59)	76.29 (69.67, 82.09)	74.56 (67.30, 80.93)	66.30 (58.98, 73.09)	71.13 (64.21, 77.40)
	SPR (%) (95% CI)	100.00 (97.84, 100.00)	95.65 (91.61, 98.10)	100.00 (98.12, 100.00)	99.41 (96.75, 99.99)	93.48 (88.89, 96.59)	98.97 (96.33, 99.87)	95.86 (91.65, 98.32)	83.70 (77.55, 88.72)	96.91 (93.39, 98.86)	86.98 (80.96, 91.66)	75.54 (68.68, 81.57)	87.11 (81.57, 91.48)
	GMT (1:) (95% CI)	852.84 (709.04, 1025.80)	746.89 (579.46, 962.70)	660.91 (574.17, 760.77)	712.02 (587.93, 862.30)	640.00 (506.46, 808.75)	489.56 (419.08, 571.88)	125.61 (107.26, 147.11)	117.92 (97.50, 142.63)	177.47 (154.23, 204.21)	87.20 (74.15, 102.54)	93.01 (76.18, 113.56)	130.05 (109.92, 153.87)

**Table 3 vaccines-11-01586-t003:** Immunogenicity comparison between three groups for each strain 28 days after the last vaccination (per-protocol set).

Strain	Comparison Group	GMT Ratio (95%CI)	*p* Value	SCR Difference (95% CI)	*p* Value	SPR Difference (95% CI)	*p* Value
H1N1	Group 1 vs Group 2	1.14 (0.83, 1.57)	0.4118	4.01 (−1.14, 9.41)	0.1195	4.35 (2.07, 8.35)	0.0076
	Group 1 vs Group 3	1.29 (1.03, 1.62)	0.0286	9.26 (3.58, 15.30)	0.0022	0.00 (−2.23, 1.95)	1.0000
	Group 2 vs Group 3	1.13 (0.85, 1.50)	0.3999	5.25 (−1.07, 11.69)	0.1009	−4.35 (−8.35, −2.22)	0.0029
H3N2	Group 1 vs Group 2	1.11 (0.82, 1.51)	0.4913	12.18 (4.45, 19.92)	0.0023	5.93 (2.46, 10.55)	0.0031
	Group 1 vs Group 3	1.45 (1.14, 1.86)	0.0027	27.49 (19.06, 35.66)	<0.0001	0.44 (−2.33, 3.16)	1.0000
	Group 2 vs Group 3	1.31 (0.99, 1.73)	0.0582	15.32 (6.06, 24.34)	0.0012	−5.49 (−10.15, −1.92)	0.0047
BY	Group 1 vs Group 2	1.07 (0.83, 1.37)	0.6179	13.71 (5.65, 21.73)	0.0010	12.16 (6.11, 18.68)	0.0002
	Group 1 vs Group 3	0.71 (0.57, 0.87)	0.0013	11.88 (4.03, 19.63)	0.0034	−1.05 (−5.57, 3.03)	0.5915
	Group 2 vs Group 3	0.66 (0.53, 0.84)	0.0007	−1.83 (−10.57, 6.86)	0.6793	−13.21 (−19.57, −7.61)	<0.0001
BV	Group 1 vs Group 2	0.94 (0.72, 1.21)	0.6242	8.25 (−1.32, 17.64)	0.0902	11.44 (3.32, 19.51)	0.0062
	Group 1 vs Group 3	0.67 (0.53, 0.85)	0.0009	3.42 (−5.83, 12.51)	0.4652	−0.13 (−7.31, 6.83)	0.9704
	Group 2 vs Group 3	0.72 (0.55, 0.93)	0.0114	−4.83 (−14.15, 4.52)	0.3111	−11.57 (−19.49, −3.77)	0.0038

**Table 4 vaccines-11-01586-t004:** The comparison of one dose or two doses of IIV4 in individuals who had a one-dose vaccination history against influenza.

Time	Indicator	H1N1			H3N2			BY			BV		
		SG 1_1	SG 2_1	*p* Value	SG 1_1	SG 2_1	*p* Value	SG 1_1	SG 2_1	*p* Value	SG 1_1	SG 2_1	*p* Value
	N	54	50		54	50		54	50		54	50	
Pre	SPR (%)(95% CI)	59.26 (45.03, 72.43)	68.00 (53.30, 80.48)	0.3550	75.93 (62.36, 86.51)	76.00 (61.83, 86.94)	0.9930	29.63 (17.98, 43.61)	38.00 (24.65, 52.83)	0.3668	18.52 (9.25, 31.43)	26.00 (14.63, 40.34)	0.3584
	GMT (1:) (95% CI)	51.05 (33.04, 78.88)	66.81 (42.83, 104.21)	0.3876	89.80 (59.66, 135.15)	97.14 (65.11, 144.92)	0.7838	20.26 (17.02, 24.12)	25.67 (20.17, 32.67)	0.1096	16.71 (12.88, 21.68)	19.19 (13.80, 26.66)	0.5071
Post	SCR (%) (95% CI)	88.89 (77.37, 95.81)	92.00 (80.77, 97.78)	0.7433	77.78 (64.40, 87.96)	74.00 (59.66, 85.37)	0.6524	85.19 (72.88, 93.38)	82.00 (68.56, 91.42)	0.6607	59.26 (45.03, 72.43)	66.00 (51.23, 78.79)	0.4780
	SPR (%) (95% CI)	100.00 (93.40, 100.00)	100.00 (92.89, 100.00)	1.0000	98.15 (90.11, 99.95)	100.00 (92.89, 100.00)	1.0000	92.59 (82.11, 97.94)	98.00 (89.35, 99.95)	0.3646	85.19 (72.88, 93.38)	88.00 (75.69, 95.47)	0.6743
	GMT (1:) (95% CI)	600.21 (437.27, 823.87)	930.55 (640.33, 1352.30)	0.0739	682.42 (474.94, 980.56)	798.93 (616.47, 1035.40)	0.4855	135.41 (98.10, 186.92)	196.98 (145.01, 267.59)	0.0948	86.40 (64.90, 115.04)	113.14 (82.03, 156.03)	0.2102

SG 1_1: a subgroup of Group 1, refers to participants who had one shot vaccination history before enrollment in Group 1; SG 2_1: a subgroup of Group 2, refers to participants who had one shot vaccination history before enrollment in Group 1. Both SG 1_1 and SG 2_1 received one dose of flu vaccine prior to this clinical trial.

**Table 5 vaccines-11-01586-t005:** The comparison of immunogenicity induced by one dose of IIV4 in individuals with a one-dose, two-dose, or no vaccination history against influenza.

**Time**	**Indicator**	**H1N1**				**H3N2**			
		**SG 2_0**	**SG 2_1**	**G3**	***p* Value**	**SG 2_0**	**SG 2_1**	**G3**	***p* Value**
	**N**	**134**	**50**	**194**		**134**	**50**	**194**	
Pre	SPR (%) (95% CI)	42.54 (34.04, 51.37)	68.00 (53.30, 80.48)	69.59 (62.59, 75.97)	<0.0001	60.45 (51.64, 68.78)	76.00 (61.83, 86.94)	77.84 (71.33, 83.47)	0.0021
	GMT (1:) (95% CI)	21.06 (16.93, 26.20)	66.81 (42.83, 104.21)	64.33 (52.72, 78.50)	<0.0001	42.34 (33.38, 53.71)	97.14 (65.11, 144.92)	99.84 (82.66, 120.59)	<0.0001
Post	SCR (%) (95% CI)	91.79 (85.79, 95.83)	92.00 (80.77, 97.78)	86.60 (80.98, 91.05)	0.2601	78.36 (70.42, 85.00)	74.00 (59.66, 85.37)	61.86 (54.62, 68.72)	0.0046
	SPR (%) (95% CI)	94.03 (88.58, 97.39)	100.00 (92.89, 100.00)	100.00 (98.12, 100.00)	0.0004	91.04 (84.88, 95.29)	100.00 (92.89, 100.00)	98.97 (96.33, 99.87)	0.0005
	GMT (1:) (95% CI)	688.07 (499.18, 948.44)	930.55 (640.33, 1352.30)	660.91 (574.17, 760.77)	0.3049	589.17 (433.38, 800.94)	798.93 (616.47, 1035.40)	489.56 (419.08, 571.88)	0.0678
**Time**	**Indicator**	**BY**				**BV**			
		**SG 2_0**	**SG 2_1**	**G3**	***p* Value**	**SG 2_0**	**SG 2_1**	**G3**	***p* Value**
	**N**	**134**	**50**	**194**		**134**	**50**	**194**	
Pre	SPR (%) (95% CI)	14.93 (9.36, 22.11)	38.00 (24.65, 52.83)	47.94 (40.73, 55.21)	<0.0001	12.69 (7.57, 19.53)	26.00 (14.63, 40.34)	24.23 (18.38, 30.88)	0.0220
	GMT (1:) (95% CI)	15.44 (13.84, 17.23)	25.67 (20.17, 32.67)	31.26 (28.13, 34.73)	<0.0001	11.09 (9.70, 12.68)	19.19 (13.80, 26.66)	15.74 (13.76, 18.01)	0.0002
Post	SCR (%) (95% CI)	71.64 (63.21, 79.09)	82.00 (68.56, 91.42)	76.29 (69.67, 82.09)	0.3203	66.42 (57.75, 74.34)	66.00 (51.23, 78.79)	71.13 (64.21, 77.40)	0.5979
	SPR (%) (95% CI)	78.36 (70.42, 85.00)	98.00 (89.35, 99.95)	96.91 (93.39, 98.86)	<0.0001	70.90 (62.43, 78.42)	88.00 (75.69, 95.47)	87.11 (81.57, 91.48)	0.0004
	GMT (1:) (95% CI)	97.38 (77.48, 122.38)	196.98 (145.01, 267.59)	177.47 (154.23, 204.21)	<0.0001	86.45 (67.49, 110.75)	113.14 (82.03, 156.03)	130.05 (109.92, 153.87)	0.0183

SG 2_0: a subgroup of Group 2, refers to participants who did not receive influenza vaccines before enrollment in Group 2; SG 2_1: a subgroup of Group 2, refers to participants who received one shot of influenza vaccines before enrollment in Group 2; G3: Group 3. Participants in SG 2_0, SG 2_1, and G3 received one dose of IIV4 in this clinical trial.

**Table 6 vaccines-11-01586-t006:** Overall vaccine-related adverse events (adverse reactions) experienced after vaccination (safety set).

AEs	Group 1, Dose 1 (N = 200)	Group 1, Dose 2 (N = 188)	Group 2 (N = 200)	Group 3 (N = 200)
Overall reactions	23 (11.50)	7 (3.72)	25 (12.50)	26 (13.00)
Grade 1	18 (9.00)	3 (1.60)	19 (9.50)	17 (8.50)
Grade 2	6 (3.00)	4 (2.13)	8 (4.00)	19 (9.50)
Grade 3 and above	0 (0.00)	0 (0.00)	1 (0.50)	0 (0.00)
General disorders and administration site conditions	14 (7.00)	3 (1.60)	17 (8.50)	25 (12.50)
Grade 1	10 (5.00)	2 (1.06)	14 (7.00)	17 (8.50)
Grade 2	4 (2.00)	1 (0.53)	5 (2.50)	18 (9.00)
Grade 3 and above	0 (0.00)	0 (0.00)	1 (0.50)	0 (0.00)
Vaccination site erythema	3 (1.50)	0 (0.00)	4 (2.00)	15 (7.50)
Grade 1	1 (0.50)	0 (0.00)	3 (1.50)	3 (1.50)
Grade 2	2 (1.00)	0 (0.00)	1 (0.50)	12 (6.00)
Pyrexia	5 (2.50)	3 (1.60)	11 (5.50)	4 (2.00)
Grade 1	4 (2.00)	2 (1.06)	7 (3.50)	0 (0.00)
Grade 2	1 (0.50)	1 (0.53)	3 (1.50)	4 (2.00)
Grade 3 and above	0 (0.00)	0 (0.00)	1 (0.50)	0 (0.00)
Vaccination site pain	4 (2.00)	0 (0.00)	3 (1.50)	6 (3.00)
Grade 1	4 (2.00)	0 (0.00)	3 (1.50)	5 (2.50)
Grade 2	0 (0.00)	0 (0.00)	0 (0.00)	1 (0.50)
Vaccination site induration	2 (1.00)	0 (0.00)	2 (1.00)	7 (3.50)
Grade 1	1 (0.50)	0 (0.00)	1 (0.50)	5 (2.50)
Grade 2	1 (0.50)	0 (0.00)	1 (0.50)	2 (1.00)
Vaccination site swelling	1 (0.50)	0 (0.00)	3 (1.50)	7 (3.50)
Grade 1	0 (0.00)	0 (0.00)	1 (0.50)	2 (1.00)
Grade 2	1 (0.50)	0 (0.00)	2 (1.00)	5 (2.50)
Vaccination site pruritus	0 (0.00)	0 (0.00)	2 (1.00)	7 (3.50)
Grade 1	0 (0.00)	0 (0.00)	2 (1.00)	7 (3.50)
Vaccination site rash	1 (0.50)	0 (0.00)	1 (0.50)	0 (0.00)
Grade 1	1 (0.50)	0 (0.00)	1 (0.50)	0 (0.00)
Grade 2	0 (0.00)	0 (0.00)	0 (0.00)	0 (0.00)
Asthenia	1 (0.50)	1 (0.53)	0 (0.00)	0 (0.00)
Grade 1	1 (0.50)	0 (0.00)	0 (0.00)	0 (0.00)
Grade 2	0 (0.00)	1 (0.53)	0 (0.00)	0 (0.00)
Fatigue	1 (0.50)	1 (0.53)	0 (0.00)	0 (0.00)
Grade 1	1 (0.50)	0 (0.00)	0 (0.00)	0 (0.00)
Grade 2	0 (0.00)	1 (0.53)	0 (0.00)	0 (0.00)
Respiratory, thoracic and mediastinal disorders	11 (5.50)	4 (2.13)	8 (4.00)	5 (2.50)
Grade 1	8 (4.00)	0 (0.00)	5 (2.50)	1 (0.50)
Grade 2	3 (1.50)	4 (2.13)	3 (1.50)	4 (2.00)
Cough	7 (3.50)	3 (1.60)	6 (3.00)	4 (2.00)
Grade 1	4 (2.00)	0 (0.00)	4 (2.00)	1 (0.50)
Grade 2	3 (1.50)	3 (1.60)	2 (1.00)	3 (1.50)
Rhinorrhoea	4 (2.00)	1 (0.53)	2 (1.00)	0 (0.00)
Grade 1	4 (2.00)	0 (0.00)	1 (0.50)	0 (0.00)
Grade 2	0 (0.00)	1 (0.53)	1 (0.50)	0 (0.00)
Oropharyngeal pain	0 (0.00)	0 (0.00)	0 (0.00)	1 (0.50)
Grade 2	0 (0.00)	0 (0.00)	0 (0.00)	1 (0.50)
Gastrointestinal disorders	1 (0.50)	1 (0.53)	2 (1.00)	1 (0.50)
Grade 1	1 (0.50)	0 (0.00)	2 (1.00)	1 (0.50)
Grade 2	0 (0.00)	1 (0.53)	0 (0.00)	0 (0.00)
Vomiting	1 (0.50)	1 (0.53)	0 (0.00)	1 (0.50)
Grade 1	1 (0.50)	0 (0.00)	0 (0.00)	1 (0.50)
Grade 2	0 (0.00)	1 (0.53)	0 (0.00)	0 (0.00)
Nausea	0 (0.00)	0 (0.00)	1 (0.50)	0 (0.00)
Grade 1	0 (0.00)	0 (0.00)	1 (0.50)	0 (0.00)
Diarrhoea	0 (0.00)	0 (0.00)	1 (0.50)	0 (0.00)
Grade 1	0 (0.00)	0 (0.00)	1 (0.50)	0 (0.00)
Metabolism and nutrition disorders	0 (0.00)	1 (0.53)	0 (0.00)	0 (0.00)
Grade 2	0 (0.00)	1 (0.53)	0 (0.00)	0 (0.00)
Decreased appetite	0 (0.00)	1 (0.53)	0 (0.00)	0 (0.00)
Grade 2	0 (0.00)	1 (0.53)	0 (0.00)	0 (0.00)
Nervous system disorders	0 (0.00)	1 (0.53)	0 (0.00)	0 (0.00)
Grade 2	0 (0.00)	1 (0.53)	0 (0.00)	0 (0.00)
Headache	0 (0.00)	1 (0.53)	0 (0.00)	0 (0.00)
Grade 2	0 (0.00)	1 (0.53)	0 (0.00)	0 (0.00)
Skin and subcutaneous tissue disorders	1 (0.50)	1 (0.53)	0 (0.00)	0 (0.00)
Grade 1	1 (0.50)	1 (0.53)	0 (0.00)	0 (0.00)
Mucocutaneous rash	1 (0.50)	1 (0.53)	0 (0.00)	0 (0.00)
Grade 1	1 (0.50)	1 (0.53)	0 (0.00)	0 (0.00)

## Data Availability

The data presented in this study are available upon request from the corresponding authors. The data are not publicly available due to restrictions concerning privacy or ethical considerations.

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
