# Peer review of "Immunogenicity and Safety of One versus Two Doses of Quadrivalent Inactivated Influenza Vaccine (IIV4) in Vaccine-Unprimed Children and One Dose of IIV4 in Vaccine-Primed Children Aged 3–8 Years"

_vaccines, 2023, doi:10.3390/vaccines11101586_

Round 1

Reviewer 1 Report

In this manuscript “Immunogenicity and Safety of One Versus Two Doses of Quadrivalent Inactivated Influenza Vaccine (IIV4) in Vaccine-unprimed, and One Dose of IIV4 in Vaccine-primed Children Aged 3-8 Years” by Yunfeng Shi et al, authors have carried out a clinical study to evaluate the efficacy of a IIV4 against flu in different groups of children aged 3-8 years. For that three groups were included in the clinical study: Children that received two-dose (Group 1) or one-dose (Group 2) regimen of IIV4, and an additional group tvaccine-primed that received one dose of IIV4 (Group 3). All vaccines used for this trial were manufactured by Sinovac Biotech Co., LTD, therefore the conclusion of the study are mainly related to this vaccine, although results could be applied to other 4IIV with similar composition. Authors have also examined the immune responses induced after vaccination. Overall, this is a well written and interesting document. While, I am agree with the data interpretation of the study it is unclear if the number of subjects (around 200 per group) is enough to obtain data with a strong statistical power. Some of the differences observed between groups were too low, which can damper the overall message of the manuscript. Supplementary tables 1 and 2 should be included in the main text.

Author Response

Dear Reviewer,

Thank you for your encouraging comments and professional suggestions. As you suggested, supplementary tables 1 and 2 has been included in the main text and renamed properly (See Table 4 and Table 5).

Reviewer 2 Report

The authors compared vaccine efficacy of quadrivalent flu vaccine (IIV4) in children aged 3-8 years old in phase IV clinical trial in China. The manuscript is well-written and easy to understand. The methodology and conclusion looks correct. Comments for the authors below:

 Major points:

1.      Line 1.8: Please describe the method of micro- hemagglutination inhibition test.

 Minor points:

1.      Line 234: “Hongkong” should be “Hong Kong”.

Author Response

Dear Reviewer,

Thank you for your comments. We have made modifications as you suggested. Please check below.

 Major points:

1. Line 1.8: Please describe the method of micro- hemagglutination inhibition test.

Response: Thank you for your advice. Description of hemagglutination inhibition test has been added in 2.7 Immunogenicity evaluation (line 107-114) as follows.

Antibody titers were measured using hemagglutination inhibition (HI) assay performed at China National Institutes for Food and Drug Control. Non-specific inhibitors were eliminated using receptor-destroying enzyme (cholera filtrate, Sigma) at 37°C for 16-18h, and spontaneous agglutinins were adsorbed with Rooster red blood cells. Starting at 1:10, serum samples were 2-fold diluted for 10 times, and incu-bated with an influenza virus solution of 4 hemagglutination units/25μL. The highest sera dilution at which hemagglutination was completely inhibited was reported as hemagglutination inhibition titer. A repeated independent HI assay was run to ensure the accuracy.

Minor points:

1. Line 234: “Hongkong” should be “Hong Kong”.

Response: Thanks for your careful check. I am sorry for this spelling typo. We have revised the related sentence as follows (line 253): “A study in Hong Kong reported that….”

Reviewer 3 Report

Sinovac Biotech Co. LTD's quadrivalent inactivated influenza vaccine (IIV4) was licensed for influenza prevention in China in 2020 and is recommended for children as young as three years old. IIV4 immunogenicity and safety results were satisfactory in a pivotal phase I/III clinical trial. However, only the one-dose regimen of IIV4 was evaluated, and the immune response in children with varying vaccination histories was not studied separately. The authors of this study assessed the need for a second dose of quadrivalent inactivated influenza vaccine (IIV4) in children aged 3 to 8 years, with a focus on vaccine-unprimed and vaccine-primed children. The study addresses an important question by taking into account the age group and potential differences in vaccine response based on prior vaccination history. It makes an important contribution to the understanding of influenza vaccination in young children. It also provides evidence to support the use of a two-dose regimen of IIV4 in unvaccinated children aged 3 to 8 years. These findings have practical implications for public health strategies relating to influenza prevention in children.

However, the study's evaluation period is limited to 28 days after vaccination. A longer-term follow-up would be beneficial for assessing the durability of immune responses, especially given the seasonal nature of influenza. Furthermore, the geographical scope of the study is limited to Huai'an County in Jiangsu, China. More diverse populations and settings should be included in future research to improve generalizability.

Author Response

Dear Reviewer,

Thank you for your valuable comments and suggestions.

The short monitoring period (4 weeks after vaccination) leads to absence of long-term immunogenicity data in this study. In future, we will consider to conduct the study of immune persistence in larger, more diverse populations to provide more evidence of immune strategies.

Reviewer 4 Report

The present study supported the recommendations of two-dose regimen for the quadrivalent inactivated influenza vaccine use to explore the optimal influenza immunization strategy for children aged 3-8 years. This is a carefully done study and the findings are of considerable interest. However, the authors made no mention of safety in the abstract. I also wonder the definition of "vaccine-unprimed". Is this a common technique for vaccine evaluation?

Minor points:

1. Group should always start with a capital letter.

2. Lines 32 and 39. Influenza virus infection or influenza not influenza infection.

3. Lines 49 and 52. Note IIV4.

4. Line 69. CDC not Center ..........

5. Lien 95. HA should be spelled out.

6. Line 96. What is influenza virus-like strain?

7. Lines 97-99. The strain name should be abbreviated here.

8. K¥Line 108. HI not hemagglutination inhibition.

9. Line 116. No sources are indicated for ref. 8 and 9.

10. Lines 122, 209, 212, and 242. Note AR.

11. Line 143. CMH-x2 should be spelled out. 

12. Lines 176 and 178. Delete post-vaccination.

13. Line 216. MedDRA should be spelled out. 

14. Line 266. Protection?

15. Lines 283-284. Immune not immunogenicity.

Use abbreviations with care.

Author Response

Dear Reviewer,

Thanks a lot for your careful reading and helpful suggestions. Our point-by-point responses are as follows.

Major Point

The present study supported the recommendations of two-dose regimen for the quadrivalent inactivated influenza vaccine use to explore the optimal influenza immunization strategy for children aged 3-8 years. This is a carefully done study and the findings are of considerable interest. However, the authors made no mention of safety in the abstract. I also wonder the definition of "vaccine-unprimed". Is this a common technique for vaccine evaluation?

Response: We appreciate your comments and question. We deemed "vaccine-unprimed" and "vaccine-primed" common technical terms in vaccine evaluation, and we also defined the terms in our manuscript (see line 67-70). The term "vaccine-unprimed" and "vaccine-primed" has been seen in many publications [1-2] and that’s why we used them in our manuscript.

Reference

[1] Claeys C, Chandrasekaran V, García-Sicilia J, et al. Anamnestic Immune Response and Safety of an Inactivated Quadrivalent Influenza Vaccine in Primed Versus Vaccine-Naïve Children. Pediatr Infect Dis J. 2019;38(2):203-210. doi:10.1097/INF.0000000000002217

[2] Stadtmauer EA, Vogl DT, Luning Prak E... Transfer of influenza vaccine-primed costimulated autologous T cells after stem cell transplantation for multiple myeloma leads to reconstitution of influenza immunity: results of a randomized clinical trial. Blood. 2011 Jan 6;117(1):63-71. doi: 10.1182/blood-2010-07-296822. Epub 2010 Sep 23. PMID: 20864577; PMCID: PMC3037760.

Question 1: The authors made no mention of safety in the abstract.

Response: Thank you for your suggestion. The safety results were provided in the first sentence of the results (line 20-22), as follows:

One-dose or two-dose of IIV4 were well tolerated and safe in children aged 3-8 years, and no se-rious adverse events related to vaccine were reported.

Minor points:

  1. Group should always start with a capital letter.

Response: Thank you for your suggestion. We have revised the group with the capital letter throughout the article.

  1. Lines 32 and 39. Influenza virus infection or influenza not influenza infection.

Response: We sincerely thank you for careful reading. We have revised “influenza infection” to “influenza virus infection” in line 33 and line 40.

  1. Lines 49 and 52. Note IIV4.

Response: Thanks for your careful checks. We have revised the sentence as follows (line 50-51):

“……who have not previously ≥2 doses of trivalent or quadrivalent inactivated influenza vaccine before July 1,2022”.

  1. Line 69. CDC not Center ..........

Response: Thanks for your careful reminder. We have checked the name in the office website, as follows: Jiangsu Provincial Center Disease Control and Prevention.

  1. Lien 95. HA should be spelled out.

Responses: Thank you for your suggestion. We have added the full name of HA (hemagglutinin) in line 94-95.

  1. Line 96. What is influenza virus-like strain?

Response: Thank you for your careful reading. We’re sorry that the previous description is not inaccurate. We have revised the sentence (line 94095), as follows:

Each 0.5 mL dose contained 15 mcg of hemagglutinin (HA) from the four influenza strains recommended……

  1. Lines 97-99. The strain name should be abbreviated here.

Response: Thank you for your suggestion. But we have reviewed many articles about the clinical trials of influenza vaccine, and all use the strain name, not the abbreviation. Therefore, we hope not to revise the strain names.

  1. Line 108. HI not hemagglutination inhibition.

Response: Thank you for your suggestion. We reviewed the articles of inactivated influenza vaccine (IIV) manufactured by Sanofi Pasteur, and GlaxoSmithKline [1-2]. The two articles and many other published articles used the term “hemagglutination inhibition” or its abbreviation “HI”. Therefore we don’t see any unreasonable use of “HI” in our manuscript. However, we are highly pleasured to know more details about your question. Thank you.

References:

[1] Greenberg DP, Robertson CA, Landolfi VA, Bhaumik A, Senders SD, Decker MD. Safety and immunogenicity of an inactivated quadrivalent influenza vaccine in children 6 months through 8 years of age. Pediatr Infect Dis J. 2014;33(6):630-636. doi:10.1097/INF.0000000000000254

[2] Kieninger, D., Sheldon, E., Lin, WY. et al. Immunogenicity, reactogenicity and safety of an inactivated quadrivalent influenza vaccine candidate versus inactivated trivalent influenza vaccine: a phase III, randomized trial in adults aged ≥18 years. BMC Infect Dis 13, 343 (2013). https://doi.org/10.1186/1471-2334-13-343

  1. Line 116. No sources are indicated for ref. 8 and 9.

We truly appreciate your advice. After verification, we have decided to delete Reference No.8 in our manuscript. The previous Reference No.8 (called Technical Guidelines for influenza vaccination in China) was a regulation issued by the Chinese National Medical Products Administration (NMPA) in 2021, which defined SPR and SCR as stated in the manuscript. The regulation has been widely adopted in the industry of influenza vaccine research and development in China, and the criteria in which were exactly the same as those in EMA’s guidance. However, after double-check, we confirm that Reference No.8 did not take effect in China. Therefore, we deleted Reference No.8.

Regarding the ref. 9, we also updated as follows:

[8] European Medicines Agency. Guideline on Influenza Vaccines-Non-clinical and clinical Module. EMA/CHMP/VWP/457259/2014, 2014.

  1. Lines 122, 209, 212, and 242. Note AR.

Thank you for your suggestion. We have noted AR for adverse reaction in line 126 and line 227.

  1. Line 143. CMH-x2 should be spelled out. 

Thank you for your advice. We have checked the statistical methods in the SAP and revised the related sentence (line 147-150), as follows:

The two-sided 95% CI of the SPRs and SCRs were calculated by using Clopper-Pearson, the Pearson χ2 test or Fisher exact test was used to compare the difference among groups and the Miettinen-Nurminen method was used to calculate the rate difference and two-sided 95% CI.

  1. Lines 176 and 178. Delete post-vaccination.

Response: Thank you for your suggestion. We have deleted the “post-vaccination” (line 183).

  1. Line 216. MedDRA should be spelled out. 

Response: Thank you for your kind reminder. We have added the full name of MedDRA in line 235.

  1. Line 266. Protection?

Response: Thank you for your kind reminder. This should be “immune protection”. (line 284)

  1. Lines 283-284. Immune not immunogenicity.

Response: Thank you for professional word suggestion. We’ve revised it to “immune response” in line 301-302.